# The Influence of the Locus of Control Construct on the Efficacy of Physiotherapy Treatments in Patients with Chronic Pain: A Systematic Review

**DOI:** 10.3390/jpm12020232

**Published:** 2022-02-07

**Authors:** Julia Álvarez-Rodríguez, Raquel Leirós-Rodríguez, Jaume Morera-Balaguer, Pilar Marqués-Sánchez, Óscar Rodríguez-Nogueira

**Affiliations:** 1Physical Therapy Section, Nursing and Physical Therapy Department, University of León, 24401 Ponferrada, Spain; jalvar08@estudiantes.unileon.es; 2SALBIS Research Group, Physical Therapy Section, Nursing and Physical Therapy Department, University of León, 24401 Ponferrada, Spain; mpmars@unileon.com (P.M.-S.); orodn@unileon.es (Ó.R.-N.); 3Department of Physiotherapy, Health Science Faculty, CEU-Cardenal Herrera University, CEU Universities, 03204 Elche, Spain; jmorera.el@uch.ceu.es

**Keywords:** locus of control, rehabilitation, chronic pain, biopsychosocial, treatment outcome

## Abstract

The biopsychosocial paradigm contemplates the patient’s personality traits in physiotherapy treatments for chronic pain. Among these traits, the locus of control has a direct relationship with the person’s coping strategies in the face of their health problems. The objective of this systematic review was to assess the influence of locus of control on the efficacy of physiotherapy treatments in patients with chronic pain. A systematic review of the publications of the last ten years in Pubmed, Scopus, Science Direct and Web of Science databases was conducting with the terms physical therapy modalities, chronic pain, internal-external control, self-management, physical therapy and physiotherapy. The inclusion criteria were participants with chronic pain lasting at least three months who have received at least one session of physical therapy; the studies should have collected the patient’s locus of control as a prognostic factor at the beginning of physiotherapy treatment; the variables studied should include the pain intensity or clinical variables related to pain. A total of 13 articles were found, of which three were experimental studies and ten were observational studies. The included samples had chronic knee pain, nonspecific back pain, low back pain or neck pain; were people over 65 years of age or patients who had undergone hand surgery. In patients with chronic pain for more than three months, the locus of control construct participates as a predictor of the results of physiotherapy treatment. The presence of an internal locus of control favors better results. The personality traits of the subjects represent an important factor to take into account when planning physiotherapy treatments.

## 1. Introduction

Pain is considered chronic when it lasts for more than three to six months and/or persists beyond the expected tissue healing time [1]. This type of pain is characterized by its variable intensity, its resistance to pharmacological treatments and the wide variety of factors that can influence its onset and maintenance [2]. The International Association for the Study of Pain defines chronic pain as persistent or recurrent pain lasting more than 3 months [1]. Chapman and Nakamura define chronic pain as “pain that is severe, long-lasting, life-altering, no longer protective, and instead impairs health and functional abilities, becoming a source of suffering and disability” [3]. Patients report that pain has a negative influence on their daily, social, occupational and emotional life, leading to changes in their behavior [4]. Given this multifactorial nature, physiotherapy (PT) has moved towards a patient-centered biopsychosocial approach in which pain is described as a product of the dynamic and multidimensional interaction between physiological, emotional, cognitive, behavioral and sociocultural factors that reciprocally influence each other [5].

Chronic pain (CP) is a symptom that affects 20–40% of the world’s population (depending on the country and age group analyzed) [6,7]. The prevalence of CP is higher with increasing age and among women [8]. The economic costs associated with CP are considerable, estimated to be over $14,000 per patient [9]. A significant proportion of the above costs are related to the prescription of drugs. However, in addition to being ineffective, painkillers are associated with significant health risks, including addiction and death [10]. Consequently, in the last decade, the recommendation has shifted towards a prioritized primary approach involving non-pharmacological interventions, such as PT [11].

Traditionally, PT has assumed that the most appropriate techniques were to be chosen according to the clinical picture, and that successful treatment was based on the implementation of these techniques [12]. The current PT paradigm takes into account the patient’s personality traits and expectations in relation to treatment [11]. Psychological factors are usually investigated and treated separately, but constructs such as depression, catastrophizing (irrational belief that something is much worse than it really is) and low self-efficacy (belief in one’s ability to influence events affecting one’s life) seem to coexist in patients with chronic pain and should be considered [13]. This new paradigm implies a shift in PT treatments towards patient education and inclusion in active exercise [14]. The need to influence the patients’ belief that they are capable of performing certain exercises involves consider the patient’s locus of control (LoC) as a variable of interest in treatment interventions and designs [15].

The LoC is a construct that defines how a person perceives events according to whether the person considers them to be a consequence of or modifiable by their own behavior (internal LoC) or by that of other people, or chance (external LoC). Consequently, in relation to health, patients with internal LoC are those who consider that their health condition is a product of their own behavior or actions, and therefore they are also individuals who are capable of taking control of the events that happen to them. Conversely, an external LoC patient considers their health condition to be the result of chance and/or other factors outside of their control [16].

Likewise, there seems to be a direct relationship between LoC and a person’s coping strategies when faced with health problems. The individual’s belief that there is nothing they can do to improve their condition produces a growing sense of threat and alarm, which facilitates the possible perpetuation of their painful state [17]. The internal LoC is a variable that favors therapeutic effectiveness in psychology, physiotherapy and multidisciplinary treatments [18].

Thus, active physiotherapy treatments, where the patient must perform therapeutic exercises in the belief that these can improve their state of health, may be influenced by the person’s LoC type. In this case, PTs should pay attention to this personality variable when scheduling certain exercises, individualizing the treatment.

Consequently, the efficacy of PT treatments in patients with chronic pain could be influenced by the patient’s LoC about their pain [19]. Therefore, the need for a systematic review was proposed with the aim of assessing the influence of LoC as prognostic factor on the efficacy of PT treatments in patients with CP, with the hypothesis that the efficacy of physiotherapy treatments in patients with chronic pain is influenced by the patient’s type of locus of control.

## 2. Materials and Methods

This study was registered on PROSPERO (ID: CRD42021230088) and followed the Preferred Reporting Items for Systematic Reviews and Meta-analyses (PRISMA) reporting guidelines and the recommendations of the Cochrane Collaboration [20,21].

The PICO question was chosen as follows: P—population: adults with chronic pain lasting at least three months and interna LoC; I—intervention: PT treatment; C—control: adults with chronic pain lasting at least three months and external LoC; O—outcome: pain intensity or clinical variables related to pain (depression, anxiety, functionality, disability, catastrophizing, avoidance of fear, self-efficacy in pain control…).; S—study designs: experimental and observational studies.

A systematic search of publications published in the last ten years was conducted throughout the month of December 2020. The search was limited to the last ten years in order to provide novel and representative information on research in this area. The databases included were PubMed, Science Direct, Web of Science and Scopus. The search strategy included different combinations with the following medical subject headings (MeSH) terms: Physical therapy modalities, Chronic pain, internal-external control and self-management. In addition, they were also combined with the following free terms: physical therapy and physiotherapy. The search strategy is presented in Table 1.

The inclusion criteria were (a) participants had to be adults with chronic pain lasting at least three months, excluding patients with chronic pain of oncological origin, systemic diseases, infectious or inflammatory diseases diagnosed with a severe psychological disorder. (b) Participants must have received at least one session of PT (ideally a full course of PT), delivered by a physiotherapist and involving some direct clinical contact. (c) The studies should have collected the patient’s LoC as a prognostic factor at the beginning of PT treatment. (d) The variables studied should include the pain intensity or clinical variables related to pain (depression, anxiety, functionality, disability, catastrophizing, avoidance of fear, self-efficacy in pain control…).

After screening the data, extracting, obtaining and screening the titles and abstracts for inclusion criteria, the full texts of the selected abstracts were obtained. The full texts of titles and abstracts lacking sufficient information regarding inclusion criteria were also obtained. Full text articles were selected by three reviewers (J.Á.-R., Ó.R.-N. and R.L.-R.), all physiotherapists with experience in conducting systematic reviews used, a data extraction form, provided that the studies met the inclusion criteria. The three authors independently collected the following data for further analysis: demographic information (title, authors, journal and year), characteristics of the sample (age, inclusion and exclusion criteria, and number of participants), study-specific parameter (study type, duration of intervention, number of sessions, techniques of physical therapy included in the intervention, follow-up and drop-out) and results obtained. Furthermore, the Oxford 2011 levels of evidence [22] and the Jadad scale [23] were used to assess the studies.

## 3. Results

### 3.1. Studies Included

Out of 662 search results, 448 studies were considered eligible for inclusion after removing duplicates. Among the 448 papers screened, 400 were excluded after abstract and title screening. Kappa score of reviewers 1 and 2 was 0.397, indicating slight agreement. Of the 48 full-text articles assessed for eligibility, 13 were finally included in the synthesis, as depicted by the PRISMA flowchart in Figure 1.

Regarding the experimental designs of the investigations analyzed, three were experimental studies [24,25,26] and ten were observational studies [27,28,29,30,31,32,33,34,35,36]. The included samples had chronic knee pain [27,34,35], nonspecific back pain [25,32], low back pain [26,31,33,36] or neck pain [24]; were of people over 65 years of age [28] or patients who had undergone hand surgery [30]. In some cases, the sample consisted of patients with CP of different aetiology and anatomical distribution [29]. Methodological characteristics of the studies analyzed are presented in Table 2.

### 3.2. Treatments Applied

Xu et al. [27], López-Olivo et al. [34] and Dhurve et al. [35] applied standard post-surgical rehabilitation treatment after total knee arthroplasty. This was similar to the treatment applied by Stewart et al. [30] after hand surgery.

In another study, manual therapy was applied (passive mobilizations) [24,33] and active exercise [24,36]. Pereira et al. [31] applied PT intervention methods (unspecified) and chiropractic techniques.

Other studies failed to report the PT treatment applied [28,33] and in other cases re-ference was only made to the application of individualized multidisciplinary treatment [25,26,29,32] which could include: PT [25,26,29,32], medication [26,29], surgical treatment [29], relaxation therapy [25,29], occupational therapy [25,26] as well as psychological interventions [25,26,29,32] (psychotherapy [26,29,32] and cognitive-behavioral techniques [25,32]). The characteristics of the interventions of the studies analyzed are presented in Table 3.

### 3.3. Assessment Tools Used

For the assessment of pain the studies used the Western Ontario and McMaster Universities Osteoarthritis Index (WOMAC) [27,34], Pain, Enjoyment of Life and General Activity (PEG) Scale [28], the Numerical Rating Scale of ten points [24,29] and the Visual Analog Scale (VAS) [25,26,30,33]. Among the studies reviewed, various pain measures were used, including a qualitative assessment of pain (mild/moderate/strong/very strong) [31], the Chronic Pain Self-Efficacy Scale [32], the McGill Pain Questionnaire [36] and The Pain Catastrophizing Scale [35].

For the evaluation of the LoC, the studies used the LoC Questionnaire [27], the Health of LoC [34], the Multidimensional Health LoC (MHLC) Scale [24,28,32,33,35,36], the German Health and Illness Related Control Beliefs Questionnaire (KKG) [29,30] (which is the result of the development and modification of the MHLC Scale), the Symptom Checklist SCL-90-R [25], the German Pain Coping Questionnaire [26] and the Beliefs about Pain Control Questionnaire [31].

### 3.4. Results Obtained

Xu et al. [27] found that patients with higher preoperative depression and anxiety had significantly more severe pain, more restricted mobility and lower functional status one year after surgery. In parallel, a greater tendency towards a preoperative internal LoC was significantly associated with lower pain intensity and higher functionality. However, a greater tendency towards a preoperative external LoC was not associated with worse outcomes in pain characteristics. Another study identified that the presence of greater external LoC was associated with a greater perception of disability caused by low back pain and a higher prevalence of depression [31]. Along the same lines as the findings by López-Olivo et al. [34] who identified that patients with lower internal LoC reported higher pain intensity and perceived disability, whereas patients with lower coping skills for problem solving and higher levels of depression reported more intense levels of pain. These fin-dings are consistent with those of Dhurve et al. [35], who identified that patients who were dissatisfied with rehabilitation treatment after total knee replacement surgery were those who reported pain after the treatment intervention and who showed significantly higher scores on the catastrophic interpretation of pain, the depression component and lower internal LoC. In the same vein, Zuercher et al. [29] identified that the application of multidisciplinary treatment resulted in a significant decrease in pain intensity in the total sample. However, patients with internal LoC showed significantly lower pain intensities at the end of the intervention period. Similarly, Farin et al. [26] identified the following as risk factors for less improvement after multidisciplinary treatment: being female, being older, having comorbidities, low pre-treatment motivation, the presence of fear-avoidance beliefs and the predominance of external LoC.

Another observational study [28] identified the association between external LoC with perceived pain intensity, reporting that the presence of internal LoC reduced the likelihood of moderate pain by 30% and severe pain by 50%. Keedy et al. [32] found that those patients who showed greater internal LoC also had greater self-efficacy in pain control and showed the greatest improvement in their physical abilities after the intervention. Oliveira et al. [33] observed that those patients who had received manual therapy treatment showed higher values of external LoC compared to patients with the same characteristics who were still on the waiting list. This finding was endorsed by Batista et al. [36] whose research concluded that patients undergoing active treatment for chronic low back pain believe they are responsible for their condition (higher internal LoC). Linden et al. [25] evaluated the efficacy of cognitive behavioral therapy compared to conventional PT treatment for chronic back pain. The authors identified that both interventions achieved similar improvements in pain-related variables but not in perceived disability. In parallel, they found that mental health characteristics were not related to outcomes in either intervention group.

Finally, another study found no significant association between LoC and pain intensity after hand surgery [30] and Groeneweg et al. [22] also failed to identify statistically significant associations between the two variables. Although they did identify statistically significant associations between credibility expectations, fear-avoidance beliefs and pain at the end of PT treatment.

## 4. Discussion

The aim of this review was to assess the influence of LoC as prognostic factor on the efficacy of PT treatments in patients with CP. The results obtained indicate that, irrespective of the treatment modalities chosen, LoC could act as a mediator in reducing pain intensity.

Most of the studies reviewed identified a direct relationship between high levels of internal LoC and lower pain intensity than subjects with inverse LoC characteristics [27,28,29,31,32,34,35]. This phenomenon could be due to the fact that individuals with high levels of internal LoC have the belief that they possess effective coping strategies to deal with their pain situation [37]. Other variables analyzed and relationships identified are the association between the predominance of internal LoC and greater improvement in pain intensity and degree of depression after PT treatment [30,31,32,35]. Furthermore, there also seems to be an inverse relationship between the level of internal LoC and perceived disability [24,25,31,32,33] and suffering from pain [31,35]. This could be because feelings of catastrophism and helplessness in the face of adversity (associated with an external LoC) are factors that mediate the relationship between chronic pain, depression and disability [19].

The majority of PT interventions applied were successful for the reduction of pain intensity, degree of depression and perceived degree of disability, immediately after treatment [24,25,29,32], in the mid-term (more than three months after completion of treatment) [30] and in the long term (two years later) [24,35]. Meanwhile, one study identified that receiving a fully passive PT treatment (passive manual therapy) increased the external LoC in patients [33]. Hence, the PT methods that can be applied to modify the LoC most likely include active treatment modalities, among which physical exercise is one of the most important [24,25,32,33,36,37]. One of the many benefits of physical activity is its ability to transform the patient into an active element, with the autonomy to influence the effects of treatment and the capacity to improve their pathology [38].

In the field of PT in patients with PC, high levels of anxiety and/or depression predispose to higher levels of disability and poorer quality of life [39]. These associations are supported by one of the studies, which identified that these relationships are favored by an external LoC [31]. These phenomena highlight the complex relationships and consequences between LoC, personality traits and patient attitudes. One such relationship identified by Stewart et al. [30] was that patients with external LoC associated with health professionals reported a greater reduction in pain intensity for the duration of the treatment (a reduction that did not remain over time once the intervention was over). A finding that seems to be consistent with previous observations is that the perception and expectations of patients and the physiotherapists themselves about the treatment seem to influence treatment outcomes, especially when there is a close relationship between physiotherapist and patient [40]; and that those patients who frequently attend a PT consultation have lower internal LoC values (compared to patients with similar clinical pictures but who consider that they do not need PT treatment) [41]. Consequently, it is plausible that the predominance of external LoC favors improvements in pain in patients who tend to suffer from a certain degree of dependency on healthcare professionals [41].

Whatever treatment modality is chosen, the ultimate goal is that it should be effective for the patient, avoiding side effects and unnecessary financial costs [42]. Consequently, the predictive character of LoC on treatment efficacy and patients’ self-efficacy and expectations on outcomes are variables to be considered when defining therapeutic goals and designing a PT treatment plan in patients with CP [24,26,29,31,32]. In this manner, health professionals in general, and physiotherapists in particular, could develop an individualized treatment approach that incorporates specific strategies to improve pain management in patients with pain management difficulties [43]. An example of this is the study by Keedy et al. [32] who observed that the achievement of beneficial results at the end of treatment is related to the decrease in external LoC levels, by chance.

In relation to the previous hypothesis that the efficacy of physical therapy treatments in patients with chronic pain is influenced by the type of patient’s locus of control. The authors can affirm that this influence has been proven. As practical implications, it is important to bear in mind the psychosocial characteristics of CP patients when planning and implementing their treatment [44]. In patients with CP of musculoskeletal origin the results are better when there is a relationship of trust between patient and physiotherapist, as the communication between both is more effective and has the capacity to create an alliance that enables the patient to gain self-confidence through active participation in the treatment [39]. All these factors, as shown in this review, will increase patients’ self-efficacy [32] and the belief that the person can do something to change their situation, a belief that is closer to an internal LoC [39].

The present study has methodological limitations that should be acknowledged, such as the inclusion of nonexperimental studies, research that involved data collection via questionnaires completed by patients without the direct control of a researcher, and the lack of long-term follow-up and control groups in most of the results. Furthermore, due to the small number of studies in the review, the conclusions drawn should be interpreted with caution. However, we must recognize the strengths of this research: such as that it is the first systematic review of the literature with this objective and that it covers research from the last ten years (which implies a wide space of time).

Further research is necessary, with experimental designs and including a long-term follow-up after the intervention period, assessing the study variables with sensitive, objective, quantitative and reliable instruments (handled by trained researchers to avoid extraneous variables) and considering the attitudes and beliefs of the physiotherapist [45] (a variable ignored by all the research analyzed).

## 5. Conclusions

In patients with CP of more than three months’ duration, the LoC construct serves as a predictor value of PT treatment outcomes. The presence of a predominant internal LoC prior to the start of PT treatment favors improved outcomes, and vice versa.

There is a relationship between LoC and pain-related variables, such as pain intensity and the degree of depression and disability.

The personality traits of the subjects represent an important factor to consider when planning treatments for the management of patients with CP, therefore the psychosocial sphere must be considered in all cases. Therefore, physiotherapists and other healthcare professionals involved in the treatment of patients with CP should consider the psychosocial characteristics of their patients when planning and implementing their treatment.

## Figures and Tables

**Figure 1 jpm-12-00232-f001:**
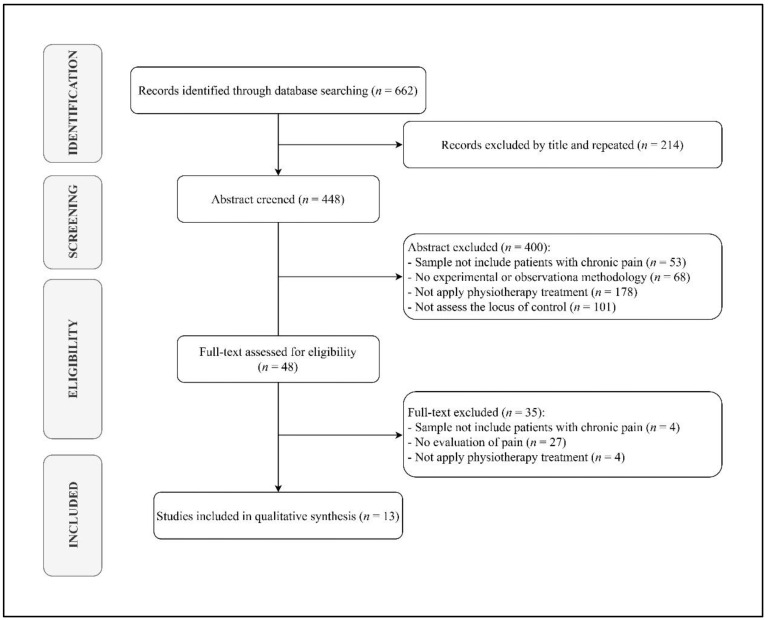
PRISMA flow gram diagram.

**Table 1 jpm-12-00232-t001:** Search equations used. MH—medical heading.

Database	Search Equation
Scopus	(MH “Physical therapy modalities”) AND (MH “Chronic pain”) AND (MH “Internal-external control”) (MH “Physical therapy modalities”) AND (MH “Chronic pain”) AND (MH “Self-management”) (MH “Chronic pain”) AND (MH “Internal-external control”) AND “Physical therapy” (MH “Chronic pain”) AND (MH “Internal-external control”) AND “Physiotherapy” (MH “Chronic pain”) AND (MH “Self-management”) AND “Physical therapy” (MH “Chronic pain”) AND (MH “Self-management”) AND “Physiotherapy”
Pubmed	(“Physical therapy modalities”(Mesh)) AND (“Chronic pain”(Mesh)) AND (“Internal-external control”(Mesh)) (“Physical therapy modalities”(Mesh)) AND (“Chronic pain”(Mesh)) AND (“Self-management”(Mesh)) (“Chronic pain”(Mesh)) AND (“Internal-external control”(Mesh)) AND “Physical therapy” (“Chronic pain”(Mesh)) AND (“Internal-external control”(Mesh)) AND “Physiotherapy” (“Chronic pain”(Mesh)) AND (“Self-management”(Mesh)) AND “Physical therapy” (“Chronic pain”(Mesh)) AND (“Self-management”(Mesh)) AND “Physiotherapy”
Web of Science	TOPIC: (“Physical therapy modalities”) AND TOPIC: (“Chronic pain”) AND TOPIC: (“Internal-external control”) TOPIC: (“Physical therapy modalities”) AND TOPIC: (“Chronic pain”) AND TOPIC: (“Self-management”) TOPIC: (“Chronic pain”) AND TOPIC: (“Internal-external control”) AND TOPIC: (“Physical therapy”) TOPIC: (“Chronic pain”) AND TOPIC: (“Internal-external control”) AND TOPIC: (“Physiotherapy”) TOPIC: (“Chronic pain”) AND TOPIC: (“Self-management”) AND TOPIC: (“Physical therapy”) TOPIC: (“Chronic pain”) AND TOPIC: (“Self-management”) AND TOPIC: (“Physiotherapy”)
Science Direct	(MH “Physical therapy modalities”) AND (MH “Chronic pain”) AND (MH “Internal external locus of control”) (MH “Physical therapy modalities”) AND (MH “Chronic pain”) AND (MH “Self management”) (MH “Chronic pain”) AND (MH “Internal external locus of control”) AND (MH “Physical therapy” OR “Physiotherapy” OR “Rehabilitation”) (MH “Chronic pain”) AND (MH “Self management”) AND (MH “Physical therapy” OR “Physiotherapy” OR “Rehabilitation”)

**Table 2 jpm-12-00232-t002:** Methodological characteristics of the studies analyzed.

Authors	Design	Sample Size	Inclussion Criteria	Exclussion Criteria	JADAD Scale				LE
					RD *	BD **	WD ***	FS	
Groeneweg et al. (2017)	RCT	181	Patients included were aged 18–70, with nonspecific subacute and chronic neck pain, with or without radiation to the shoulder region or the upper extremities, and with or without headache	Presence of red flags, pregnancy, whiplash trauma as cause, and treatment for neck pain in the previous three months	1	0	1	2	1
Pereira et al. (2017)	RCT	338	Age between 18 and 65, a diagnosis of common chronic lower back pain for a period of more than three months being attributed to muscle ligaments and mechanical and degenerative causes (according to the diagnostic criteria defined by the Portuguese Association of Rheumatology	Critical limitation on movement or diagnosis of severe psychiatric illness according to the patient’s medical chart	0	0	0	0	2
de Souza et al. (2015)	CSS	28	Age between 18 and 55 years, presenting low back pain for more than three months, currently undergoing active treatment with low back stabilization exercises and educational guidance with emphasis on self-treatment and control of their health condition	Patients presenting specific diagnosis for low back pain, such as tumors, trauma, infections, inflammatory disorders and motor and/or cognitive neurological deficit, nor being pregnant or in six months or less of postpartum	0	0	0	0	2
Keedy et al. (2014)	RS	61	Patients completing the two-week chronic spine rehabilitation program involving an interdisciplinary treatment approach including physical therapy, cognitive-behavioral group therapy, vocational rehabilitation, and group discussions with a physiatrist. Participants were at least 18 years old and English-speaking.	Not specified	0	0	0	0	2
Linden et al. (2017)	RCT	103	Patients were suffering from back pain for at least six months according to the medical records and the assessment of the treating physicians.	Patients excluded if they were applying for early retirement	1	0	1	2	1
Oliveira et al. (2012)	CSS	100	Patients with symptoms of nonspecific LBP, with symptom duration of 3 months and over, between 18 and 60 years old and being treated or awaiting treatment with a physical therapist for LBP.	Patients with fracture, tumor, infectious or inflammatory diseases of the spine and sciatica	0	0	0	0	2
Musich et al. (2020)	CSS	3824	Patients over 65 years of age with a minimum of 12 months’ continuous medical and drug plan enrollment with back pain, osteoarthritis or rheumatoid arthritis	Patients with cancer, trauma or drug abuse	0	0	0	0	2
López-Olivo et al. (2011)	PS	241	Patients with radiological diagnosis of knee osteoarthritis; first knee replacement (previous hip replacement was allowed); adequate cognitive status; living in the community (not in long-term care facilities) and with ability to communicate in English	Patients in revision surgery; with inflammatory arthropathies; neurological disorders; Paget’s syndrome or bone disorders; litigation process related to surgery and patients seeking or receiving workers’ compensation benefits.	0	0	1	1	2
Farin et al. (2011)	PS	668	Patients with chronic lower back pain for at least 6 months	Patients with specific low back pain due to tumors or inflammatory diseases	0	0	0	0	2
Xu et al. (2020)	CSS	136	Patients over 18 years old, primary unilateral or bilateral total knee arthroplasty, and English speakers	Previous septic joint, revision surgery, dementia, or were unable to return for all extra follow-up visits.	0	0	0	0	2
Zuerche-Huerlimn et al. (2019)	RS	225	Patients with somatoform pain disorder or suffering from a comorbid chronic pain condition with a mental or behavioral disorder confirmed by a clinician	Not specified	0	0	0	0	2
Stewart et al. (2018)	PS	125	Patients admitted to a tertiary hand surgery center with at least 18 years old	No selection criteria were implemented regarding pain levels at entry or comorbid diagnoses	0	0	1	1	2
Dhurve et al. (2017)	CSS	301	Patients underwent a primary unilateral total knee replacement using computer navigation, operated by two consultant orthopedic surgeons with a follow-up period ranging from one to five years.	Patients with bilateral total knee replacement or revision cases were excluded.	0	0	0	0	2

RCT: randomized controlled trial; RS: retrospective study; CSS: cross-sectional study; FS: final score; LE: level of evidence. * RD: randomization (one point if randomization is mentioned; two points if the method of randomization is appropriate). ** BD: blinding (one point if blinding is mentioned; two points if the method of blinding is appropriate). *** WD: withdrawals (one point if the number and reasons in each group are stated).

**Table 3 jpm-12-00232-t003:** Characteristics of the interventions of the studies analyzed.

Authors	Intervention		Assessment Tools	Outcomes
	Experimental Group	Control Group		
Xu et al. (2020)	Not described	---	Western Ontario and McMaster Universities Osteoarthritis Index (WOMAC), Medical Outcomes Study SF-12 – Mental Score, Hospital Anxiety and Depression Scales and LoC Questionnaire	Higher scores in preoperative depression and anxiety worse WOMAC score at 6 and 18 months. Low SF-12 score worst total WOMAC score at 6 weeks. Highest internal LOC less pain and better score in WOMAC at 18 weeks. Higher external LoC was not correlated with lower WOMAC scores. Patients with preoperative internal LOC, total WOMAC better at one year than internal LOC patients who switched to external
Musich et al. (2020)	Not described	---	Pain, Enjoyment and General Activity Assessment Scale (PEG), Veterans Rand 12, Patient Health Quesionnaire-2, Pittsburgh Sleep Quality Index, Multidimensional Health LoC and Six-item Brief Resilience Scale	The prevalence of internal LoC was 30%, external LoC (others) 34% and external LoC (chance or luck) 36%. The internal LOC was protective, reducing the likelihood of moderate pain by 30% and severe pain by 50%. The internal LOC was as protective of pain severity as having high resilience and diverse social networks. External LOC was associated with a 10% increase in moderate pain, while the external LOC subscale associated with luck was associated with a 50% increase in the likelihood of severe pain
Zuercher-Huerlimn et al. (2019)	Not described	---	German Health and Illness Related Control Beliefs and Numerical Rating Scale	High values of internal LoC showed less pain at the end of treatment. Internal LoC showed predictive value of decreased pain intensity
Stewart et al. (2018)	Not described	---	German Health and Illness Related Control Beliefs, Hospital Anxiety and Depression Scales and Visual Analogue Scale	Decrease in pain intensity, predominantly in subjects with severe pain. High levels of external LoC dependent on health professionals favour a decrease in pain intensity
Dhurve et al. (2017)	Not described	---	Pain Catastrophizing Scale, 21-Question Depression, Anxiety and Stress Scale, Multidimensional Health LoC, Oxford Knee Score (OKS) and Veterans Rand 12.	Persistent pain was the most common reason for dissatisfaction. Dissatisfied patients reported a significantly higher mean PCS score, higher depression component and lower internal locus of control. The dissatisfied group exhibited reduced improvement in OKS and range of movement, as well as a lower preoperative grade of osteoarthritis compared to satisfied patients
Groeneweg et al. (2017)	Passive mobilization techniques very gently and generally pain-free	Active exercises, improving strength, mobility, movement coordination, and relaxation, manual traction for pain reduction, and massage therapy for relaxation.	Credibility Expectancy Questionnaire, Multidimensional Health LoC, Fear Avoidance Belief Questionnaire, Neck Disability Index, Numeric Rating Scale on Pain, Medical Outcomes Study Short Form 36 and Global Perceived Effect	Treatment outcome expectancy predicted outcome success, in addition to clinical and demographic variables. Expectancy explained additional variance, ranging from 6% (pain) to 17% (functioning) at 7 weeks, and 8% (pain) to 16% (functioning) at 26 weeks. Locus of control and fear avoidance beliefs did not add significantly to predicting outcome
Pereira et al. (2017)	Physiotherapy treatment	Chiropractic treatment	Sociodemographic questionnaire, Beliefs about Pain Control Questionnaire, Illness Subjective Suffering Inventory, Oswestry Disability Questionnaire and Hospital Anxiety and Depression Scales	Suffering was a mediator in the relationship between depression and functional disability in both treatment groups. Only beliefs related to external chance events mediated the relationship between depression and functional disability in the physical therapy group, but not in the chiropractic treatment group
De Souza et al. (2015)	Not described	---	Oswestry Disability Index, McGill Pain Questionnaire, Multidimensional Health LoC	Participants presented a mean of 26 points scale for disability and 6.39 for pain. 82.1% of the participants presented higher rates for internal locus of control. Patients undergoing active treatment for chronic low back pain believe they are responsible for their own condition
Keedy et al. (2014)	Not described	---	Multidimensional Health LoC, Chronic Pain Self-Efficacy Scale, Medical Outcomes Study Short Form 36, Oswestry Disability Index and Beck Depression Inventory-II	Higher internal and lower doctor health locus of control, and higher self-efficacy at baseline predicted higher lift scores one month after treatment. Higher baseline self-efficacy also predicted better physical functioning and lower disability at one month
Linden et al. (2014)	Cognitive behavior group therapy for back pain	General orthopedic inpatient treatment, sport therapy and physiotherapy, balneotherapy, massages, or electrotherapy	Fear Avoidance Beliefs Questionnaire (FABQ), Visual Analogue Scale for Pain, Pain Disability Index and Symptom Checklist	In both groups there was a significant improvement in Symptom Checklist, the Rating of Health LoC Attributions, FABQ and Visual Analogue Scale for Pain. There are significant interactions between treatment group and Visual Analogue Scale for Pain and the FABQ, showing a superior improvement in the intervention group
Oliveira et al. (2012)	Patients undergoing physiotherapy treatment (at least one session)	Participants awaiting treatment recruited from waiting lists or from first consultations	Multidimensional Health LoC, Visual Analogue Scale and Roland Morris Disability Questionnaire	Health locus of control was found to be different between treatment and control groups. Participants being treated had higher external LoC and lower internal LoC than control group
López-Olivo et al. (2011)	Not described	---	Western Ontario and McMaster Universities Osteoarthritis Index (WOMAC), Knee Society Rating System (KSRS), Coping Responses to Stressors Inventory, Multidimensional Health LoC, Arthritis Self-Efficacy Scale and Life Orientation Test	Higher pain scores were associated with lower education and problem-solving skills, higher dysfunction and lower internal health LoC. Worse WOMAC scores were associated with less support, depression and decreased coping skills for problem solving. Older age, less education, depression, and less coping skills were significantly associated with lower total KSRS scores. A worse pain, range of movement, and knee stability score was predicted by lower problem-solving ability
Farin et al. (2011)	Not described	---	Perceived Involvement in Care Scale, Trust in Physician, General Patient Satisfaction, Communication Behavior Questionnaire, Visual Analogue Scale, Oswestry Disability Questionnaire, Fear Avoidance Beliefs Questionnaire, Control Beliefs Concerning Illness and Health and Illness perception questionnaire	The patient–physician relationship is significantly associated with the outcome. In the medium term (6 months after rehabilitation), the effect of the patient–physician relationship is clearer than in the short term (end of rehabilitation). In addition, risk factors for less improvement are female gender, higher age, low income, comorbidity, low treatment motivation, fear avoidance beliefs, and external locus of control. Future studies should examine the causal paths between the relationship variables and the outcome variables

LoC: locus of control.

## Data Availability

The data presented in this study are available on request from the corresponding author.

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
