# Peer review of "The Influence of the Locus of Control Construct on the Efficacy of Physiotherapy Treatments in Patients with Chronic Pain: A Systematic Review"

_jpm, 2022, doi:10.3390/jpm12020232_

Round 1

Reviewer 1 Report

The manuscript is important to the field of Physiotherapy (PT). Furthermore, it is well written and organized. The Manuscript needs some of minor modifications. Please find the following comments: 

1- Introduction:
- Please provide more about the relationship between the PT and Locus of Control 
- Please clarify what the current evidence has? what the main limitation in the current evidence that where your manuscript can help

2- Methods: 
- Please justify why the inclusion criteria were limited to the last ten years
- How the the Oxford 2011 Levels of Evidence can assess the quality of included studies? it is not true

3- Results: 
Please rewrite the ''3.4 Results obtained'' section in more consensus, informative, and short form. 

Author Response

Dear Editor and Reviewer of Journal of Personalized Medicine:

Thank you very much for your suggestions and contributions to improve the quality of the manuscript. Following your indications, we respond, point by point, to the reviewers' comments.

In the text, all the modified or added sentences have been written in red to facilitate the correction by the reviewers.

  1. Introduction: Please provide more about the relationship between the PT and Locus of Control.

We include a paragraph in the introduction to explain this relationship

  1. Introduction: Please clarify what the current evidence has? what the main limitation in the current evidence that where your manuscript can help.

We include a paragraph in the introduction to explain the mail limitation in the current evidence

  1. Methods: Please justify why the inclusion criteria were limited to the last ten years

The search was limited to the last ten years in order to provide novel and representative information on research in this area.

This explanation has been included in the text.

  1. Methods: How the Oxford 2011 Levels of Evidence can assess the quality of included studies? it is not true.

The sentence has been corrected.

  1. Results: Please rewrite the ''3.4 Results obtained'' section in more consensus, informative, and short form.

The authors have rewritten the indicated section following your recommendations.

Once again, thank you very much for the time spent and the interest shown in this work; as well as in the positive evaluations you have given of it.

Receive a warm greeting,

The authors.

Reviewer 2 Report

This systematic review aims to assess the influence of Locus of Control on the efficacy of physiotherapy treatments in patients with chronic pain. The topic is relevant and interesting. However, minor comments should be addressed;

Introduction:

The introduction does not cover all elements of the study. Describe specific chronic pain included in this review in detail.

Explain more about the rational of the study.

Add the hypothesis of the study.

Discussion:

Explain the implications and strengths of the study in line with objective, hypothesis, and findings of the study in detail.

Add the future directions.

Conclusion:

Add your recommendations "brief message" to the readers and researches. 

Author Response

Dear Editor and Reviewer of Journal of Personalized Medicine:

Thank you very much for your suggestions and contributions to improve the quality of the manuscript. Following your indications, we respond, point by point, to the reviewers' comments.

In the text, all the modified or added sentences have been written in red to facilitate the correction by the reviewers.

  1. Introduction: The introduction does not cover all elements of the study. Describe specific chronic pain included in this review in detail.

We include a paragraph in the introduction to explain chronic pain

  1. Introduction: Explain more about the rational of the study.

We include a paragraph in the introduction to explain more about rational of the study

  1. Introduction: Add the hypothesis of the study.

We add the hypothesis of the study

  1. Discussion: Explain the implications and strengths of the study in line with objective, hypothesis, and findings of the study in detail.

The authors have expanded the Discussion with the details you indicate.

  1. Discussion: Add the future directions.

The authors have expanded the Discussion with possible future lines of research.

  1. Conclusion: Add your recommendations "brief message" to the readers and researches.

The authors have expanded the Conclusions to include this detail.

Once again, thank you very much for the time spent and the interest shown in this work; as well as in the positive evaluations you have given of it.

Receive a warm greeting,

The authors.